# Cell Segmenter: A General Framework for Multi-modality Cell Segmentation

**Kaiwen Hu**
Zhejiang University
Hangzhou, 310027
kaiwenhu@zju.edu.cn

**Shengxuming Zhang**
Zhejiang University
Hangzhou, 310027
zsxm1998@zju.edu.cn

**Zhijie Jia**
Zhejiang University
Hangzhou, 310027
JiaZhiJie@zju.edu.cn

**Lechao Cheng**
Zhejiang Lab
Hangzhou, 310027
chenglc@zhejianglab.com

**Zunlei Feng**
Zhejiang University
Hangzhou, 310027
zunleifeng@zju.edu.cn

## Abstract

Cell Segmentation is an initial and fundamental step in biomedical image analysis, which strongly affects the experimental results of this analysis. Recently, deep learning based segmentation methods have shown great power in segmentation accuracy and efficiency. However, these data-driven methods still face many challenges, such as lack of annotations, multi-modality, and complex morphology , where morphological complexity significantly limits model performance. In this paper, we propose a new all-purpose framework with high morphological adaptability for multi-modality cell segmentation, termed Cell Segmenter (CS). For high convex cells with an arbitrary size, the Anchor-based Watershed Framework (AWF) precisely locates well-defined cell centers and generates segmentation based on these markers. For those elongated or non-convex cells, the center-independent segmentation method Omnipose [1] is adopted to obtain satisfying masks. In the inference time, confidence-based quality estimation is conducted on the branch predictions if needed, and then the better result is chosen as the final segmentation. The F1-score of the proposed method reaches 0.8537 on TuningSet and 0.6216 on the final test set of the NeurIPS 2022 Cell Segmentation Challenge.

## 1 Introduction

As an initial and fundamental step of biology and biomedical image analysis, the performance of cell segmentation significantly affects image-based biomedical research. In the past few years, Convolutional Neural Network (CNN) has achieved remarkable success in computer vision tasks, including semantic segmentation [2, 3, 4], image classification [5, 6], object detection [7, 8], etc. It inspires many researchers to focus on CNN-based cell segmentation.

Despite the current success of deep learning based methods in bio-informatics image analysis, they still need to overcome several challenges which strongly affect their applications. Firstly, to achieve a comparable performance based on a data-driven learning strategy, abundant and clean labeled data is essential for deep learning models' training. However, large amounts of annotations are costly and time-consuming, especially in biology data. Meanwhile, the entirely different statistics of images from varying microscopes increase the difficulty of handling all types of microscopy images in one model. Besides, the huge input size in whole slide images also places great demands on model efficiency.

36th Conference on Neural Information Processing Systems (NeurIPS 2022).

Furthermore, cell segmentation, as a challenging task to accurately identify individual cells in the input images, has many domain-specific problems to solve. Currently, based on the basic idea of their methodology, most deep learning cell segmentation solutions can be categorized into the following three types: (i) contour-aware segmentation, (ii) marker-based watershed algorithm, and (iii) gradient-flows based tracking methods.

The first category, contour-aware segmentation [9, 10], often adopts a three-class segmentation network to differentiate pixels of cell borders and bodies from the background. In the inference stage, any touched cell bodies will be split with cell border prediction to reduce under-segmentation. To achieve instance segmentation, Chen et al. [9] proposed a contour-aware framework based on a multi-level contextual FCN [2]. By regarding border pixels as a special category, they formulated nuclei/gland segmentation as a multi-task problem. Similarly, Panigraphi et al. [10] also adopted a three-class U-Net in bacterial cell segmentation, with an extra pre-processing step to make the model insensitive to intensity. However, all these contour-aware cell segmentation methods are designed for single modality or single cell line, and a robust cell boundary cannot be appropriately generated in multi-modality schemes or imprecise boundary annotations.

The marker-based watershed segmentation algorithm is a classical method for general segmentation tasks. Recently, many researchers [11, 12, 13, 14] attempted to adopt this method in cell segmentation tasks with a DNN-based watershed energy map and markers. To segment differential interference contrast (DIC) images with high accuracy, Lux et al. [12] extracted a marker map from the cell mask, in the post-processing stage, any unlabeled cell pixels will be labeled with the watershed algorithm. To make the model learns better, they also proposed a weighted map to force the model to focus more on the markers' boundary. Kucharski et al. [13] adopted a more complicated way to reduce over-segmentation in the watershed algorithm. They extracted four maps from ground truth annotations: cell boundary, cell body, cell center, and background maps. For better cell localization performance, Koyuncu et al. [11] solved the cell detection problem from a multi-task perspective. They compound three tasks in one FCN model: inner distance regression, normalized outer distance regression, and binary classification. They quantitively discovered that the two auxiliary tasks (outer distance regression and binary classification) could promote the localization accuracy of the inner distance map. Based on the predictions of their detection model, the cell mask can also be obtained with the marker-based watershed method. Greenwald et al. [14] further developed this method by adding a scale factor $\alpha$ to inner the distance map, which brought scale-irrelevant cell localization ability to the former model. However, these marker-based watershed methods are limited by the marker generation process. As the morphology of cells is diverse and complicated, center markers cannot be generated for some cells.

Gradient-flow tracking methods [15, 1] segment cells by tracking pixels in the predicted gradient fields. All pixels in the same cell will be aggregated into a small individual region or even a point so that pixels of touched cells will naturally be split. Stringer et al. [15] first combined gradient-flow tracking with DNN methods to realize general cell segmentation. They proposed a gradient field generation method named heat diffusion for cells with arbitrary morphology. By training DNN models to predict this field, they successfully segment most cells by gradient-flow tracking. However, in practice, their method fails with elongated cells, whose center often locates near or outside the cell boundary. Therefore, Cutler et al. [1] proposed several enhancements. For gradient field generation, instead of heat diffusion, they adopted the gradient vector from the solution of the eikonal equation. To suppress over-segmentation in the post-processing, they further applied the points cluster algorithm for a better result. Though these gradient-flow tracking methods can handle cells with diverse morphology, they will fail with large cells.

In the context of multi-modality cell segmentation, we expect to design an all-purpose cell segmentation framework for cells of arbitrary size and morphology. Currently, most cells can be categorized into three shapes: (i) simple cells with round or rod-like shapes; (ii) mutant cells, usually highly non-convex but not long; (iii) elongated cells. Due to their very different natures, an anchor-based segmentation framework with two-step watershed post-processing is proposed for segmenting cells with simple or mutant shapes. As for those elongated cells with small sizes, the recent morphology-independent solution Omnipose is adopted to produce high-quality masks. Because the two branches cannot handle all cell morphology individually, we further propose a test-time prediction quality estimation tool to automatically select the better prediction.

Our contributions can be summarized as follows: (i) We propose a general cell segmentation framework, termed Cell Segmenter, to better handle multi-modality data distribution in the challenge. (ii) In the AWF branch, a two-step watershed algorithm is adopted to precisely locate cell centers in the input. (iii) Experiment results on Cell Segmentation Challenge dataset show that the proposed Cell Segmenter can effectively segment various types of cells and achieve promising segmentation performance.

The remaining parts of this paper are structured as follows: Section 2 introduces all the details of the proposed pipeline, including pre-processing, network architecture, and post-processing. Section 3 provides the configurations of the training process and experiments. In Section 4, the quantitive ablation study and qualitative experiments are conducted over the provided TuningSet of the Cellseg Contest. Besides, an efficiency test is also conducted on the dataset. At the end of this section, the limitations of the framework and future work are discussed.

## 2 Method

In this section, we will introduce the detail of the Cell Segmenter, including the pre-processing, the detail of the proposed method, and the post-processing. In the method detail section, we first introduce the basic idea of our framework. Then the holistic architecture of the pipeline will be described, followed by the details of each module.

### 2.1 Pre-processing

The pre-processing process includes the following aspects: label cleaning, noise data removal, image normalization, and intermediate target generation.

**Label cleaning.** We split cell pixels without 4-connectivity, then remove all objects with < 5 pixels and relabel all cells.

**Noise data removal.** In practice, we find several images with either incorrect annotations or complex morphology in the given training dataset, which the proposed AWF cannot handle (F1-scores of these models are nearly zero with threshold 0.5). Therefore, we remove these images (image ID: 142, 143, 144, 443, 528, 529, 547, 548) with low F1-scores in the training process.

**Image normalization.** In the training and inference stages, all input images will be channel-wise normalized to $[0, 1]$.

**Intermediate target generation.** For the AWF network's training, the target energy map is synthesized as follows: (i) execute euclidean distance transform on the cell masks; (ii) normalize all values in the cell with its median distance, so that energy in the cells roughly ranges from 0 to 2, to generate a scale-invariant energy map. Here, the median distance is normalized to 1 for numerical stability. The cell-wise normalization operation avoids producing a wide numerical energy range for cells with different sizes and morphology.

### 2.2 Cell Segmenter

Early DNN-based nucleus/cell segmentation frameworks are mainly designated or optimized for one or few cell lines and cannot perform well when the datasets do not match their assumption. Therefore, to segment multi-modality cells of varied microscopy types, different sizes, and changing morphology, many works have been proposed in the past few years. However, all these methods still have limited performance in some challenging modalities, e.g., multinucleated cells and irregular morphology.

As the main obstacle to the general cell segmentation framework is the complex cell morphology, to achieve general cell segmentation, we propose a general framework, termed Cell Segmenter, as shown in Figure 1, by automatically selecting the segmentation method based on the morphology of input cells. We categorize cells into the following three classes: (i) simple shape, (ii) mutant shape, and (iii) elongated shape. For the cells of simple or mutant shape, we propose a detection-based watershed segmentation method, named Anchor-based Watershed Framework (AWF). For those elongated small cells, we adopt the most recent morphology-independent cell segmentation solution, Omnipose [1], to generate satisfying masks. In the inference stage, a quality estimator is adopted to automatically

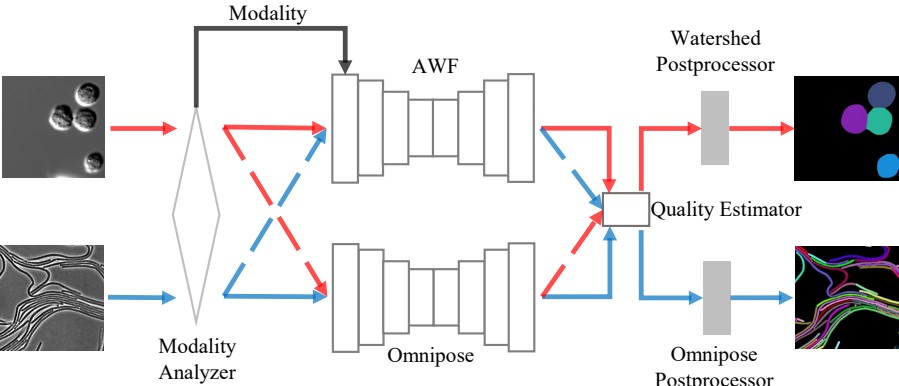

Figure 1: The whole framework of the proposed Cell Segmenter, which can be grouped into four parts: the Modality Analyzer, the Anchor-based Watershed Framework, the Omnipose Framework, and a Quality Estimator. In the pre-processing stage, the Modality Analyzer determines the input's modality based on color statistics. After that, two branches predict the intermediate representation of segmentation, and then the mask quality is computed by Quality Estimator with the branch output. Finally, better segmentation is generated by the corresponding postprocessor. The dashed arrows in the figure represent the rejected bad segmentation, while solid arrows represent the chosen results. The modality predicted by the analyzer is used for choosing the optimized AWF.

select the better prediction of the two methods. The proposed Cell Segmenter successfully solves the multi-modality issue in the cell segmentation task.

### 2.2.1 Anchor-based Watershed Framework

Marker-based Watershed segmentation, a classical segmentation algorithm widely applied in many scenarios, generates masks with pre-defined markers and an intermediate energy map. It executes flood-fill over the energy map to fill all regions which contain markers. Therefore, the quality of markers strongly affects the quality of produced segmentations. Recently, some works [11, 12, 13, 14] have attempted to combine CNN and Marker-based Watershed Transform to achieve cell segmentation. All these methods attempted to classify or regress a 'center' for each cell in a pixel-wise manner, which is only readily defined in the single-modality scenario where cells are in high homogeneity. However, in multi-modality scenarios, this pixel-wise intermediate result is practically difficult to be post-processed due to the varied cell size. Abundant datasets have verified the effectiveness of the anchor-based object detection framework on regular objects of arbitrary size. Considering the potential to generate a unique 'center' for each object based on the detection box, we propose an Anchor-based Watershed Framework (AWF) for high-convex cell segmentation of arbitrary size by combining anchor-based object detection method YOLOv5 and watershed segmentation.

As shown in Figure 2, the framework has a U-Net like structure and is mainly modified from YOLOv5-small. Unlike the classical U-Net, the AWF network has an extra detection head. In the segmentation head, the network is trained to do a foreground segmentation and a watershed energy regression. The loss function for this head defines as follows:

$$\mathcal{L}_{\text{seg}} = \mathcal{L}_{BCE}(m_{\text{pred}}, m_{\text{gt}}) + \mathcal{L}_{MSE}(e_{\text{pred}}, e_{\text{gt}}), \tag{1}$$

where $m_{\text{pred}}$, $m_{\text{gt}}$, $e_{\text{pred}}$, and $e_{\text{gt}}$ represent the output mask, the target binary mask, the output watershed energy map, and the target energy map, respectively, while $\mathcal{L}_{BCE}$ stands for the binary cross entropy loss and $\mathcal{L}_{MSE}$ stands for the mean square error loss function.

Due to the heterogeneity of cell size and density, the detection head cannot obtain valid gradients for those images with few cells. To aid this issue, we add weight factors to the objectness loss based on box amounts in each sample. Formally, for the $k$-th image $I_k$ in the batch of size $K$, it will contains $n_k$ bounding boxes, where $1 \leq k \leq K$. Then the batch-wise weight factors $w_k$ of $I_k$ can be

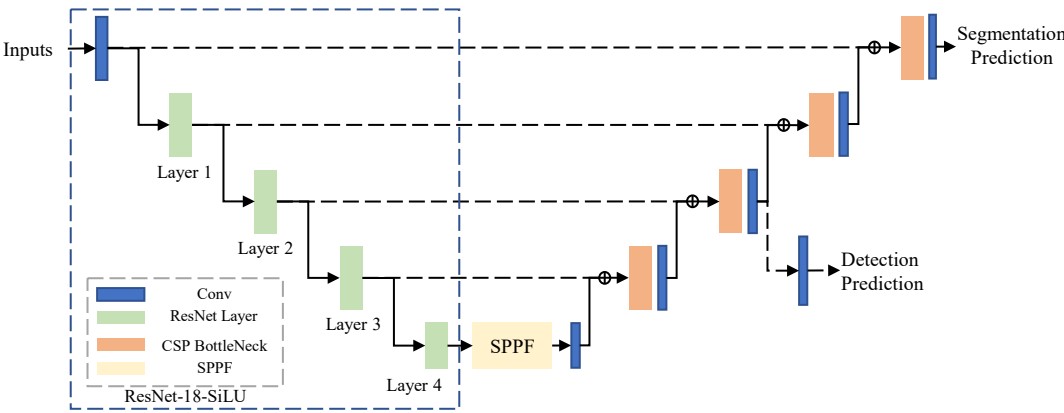

Figure 2: The network is mainly adapted from YOLOv5-small and U-Net [3]. In the encoder, we only replace ReLU with SiLU [16] of the pre-trained ResNet-18 backbone on ImageNet. The Segmentation branch will generate pixel-wise predictions, i.e., for AWF, it will predict foreground probabilities and watershed energy, while for the Omnipose model, it will compute boundary probabilities, foreground probabilities, and Omnipose flow fields. The detailed structure of each module used in the network is given in the A.1.

computed as follows:

$$w_k = \begin{cases} 0, & n_k = 0 \\ \frac{1}{n_k}, & n_k > 0 \end{cases}, \tag{2}$$

then an extra normalization is adopted over the $w_k$:

$$\tilde{w}_k = \frac{w_k}{\sum_{k=1}^{K} w_k}. \tag{3}$$

Then the re-weighted objectness loss is calculated by $\mathcal{L}_{\text{obj}} = \sum \tilde{w}_k v_k^{\text{obj}}$, where $v_k^{\text{obj}}$ is the objectness loss of the $k$-th image. Finally, the detection loss is calculated as follows:

$$\mathcal{L}_{\text{det}} = \mathcal{L}_{\text{obj}} + \alpha \mathcal{L}_{\text{loc}}, \tag{4}$$

where $\mathcal{L}_{\text{loc}}$ is a cIoU [17] loss to regress bounding boxes as YOLOv5 does, $\alpha$ is a hyperparameter to balance $\mathcal{L}_{\text{obj}}$ and $\mathcal{L}_{\text{loc}}$.

Then losses of the two heads are summed up to get the final loss:

$$\mathcal{L} = \mathcal{L}_{\text{det}} + \beta \mathcal{L}_{\text{seg}}, \tag{5}$$

where $\beta$ is also a balance factor.

### 2.2.2 Omnipose Branch

For those images AWF cannot precisely segment, e.g., the removed training data in the pre-processing step, we adopt the recently proposed Omnipose [1]. The Omnipose in our framework differs from the official implementation in the following aspects: (i) We replace the official Omnipose model with our AWF network for transfer learning; (ii) The foreground cell pixels are segmented straightforwardly from the binary prediction instead of the hysteresis threshold on the predicted Euclidean Distance Transform (EDT) map; (iii) Due to the modified learning targets of the method, the loss on the EDT map is replaced with BinaryCrossEntropy.

### 2.2.3 Quality Estimation

To automatically select the better segmentation for cells with different morphology, in the inference time, the quality of predictions from the two branches is estimated. The better one will be chosen as the final result. In detail, the confidence-based quality is estimated with the average confidence of the two branch predictions. Formally, for an input image $I$, both branches will output a map $m^{\text{AWF}}$

(or $m^{\text{Omni}}$) with pixel cell probability. Then the average foreground confidence $C(m)$ of map $m$ is computed as follows:

$$C(m) = \frac{\sum_{i=1}^{H} \sum_{j=1}^{W} m_{ij} * \mathbb{1}(m_{ij} > 0.5)}{\sum_{i=1}^{H} \sum_{j=1}^{W} \mathbb{1}(m_{ij} > 0.5)}, \tag{6}$$

where $\mathbb{1}()$ is the indicator function. The Omnipose prediction is chosen only when $C(m^{\text{AWF}}) < 0.8$ or $C(m^{\text{Omni}}) - C(m^{\text{AWF}}) > 0.05$.

## 2.3   Post-processing

In the post-processing, two enhancements are proposed to the basic marker-based watershed segmentation: (i) modality optimization; (ii) a two-step watershed algorithm.

**Modality optimization.** For better performance, the post-processing step is optimized for each modality of the input image. To achieve this, we first classify all microscopy images into grayscale, fluorescence, and brightfield. For grayscale input images, we predict masks with the grayscale-optimized AWF model and Omnipose model, then choose the better result. At the same time, we only adopt an optimized AWF model for acceleration for the other two modalities.

**Two-step watershed algorithm.** Marker-based watershed algorithm severely suffers from over-segmentation due to false cell centers, especially in multi-scale settings. Therefore, we propose a detection-based marker generation policy, which effectively reduces over-segmentation. However, the non-max suppression will fail in cell clusters, where bounding boxes of cells naturally have relatively high IoU. To solve all these problems, we design a two-step watershed algorithm. Firstly, a non max suppression is applied to the predicted bounding boxes. After NMS operation, each box will at most contain one cell and the box's center can be selected as cell marker. After the first flood-fill pass over regions with energy less than $-E_1$, the remaining unlabeled regions with $> 5$ pixels and energy lower than $-E_2$ are selected as new markers. Then the second flood-fill pass is run to obtain the final AWF segmentation.

Table 1: Development environments and requirements.

| | |
|---|---|
| System | Ubuntu 18.04.5 LTS |
| CPU | Intel(R) Xeon(R) CPU E5-2620 v4 @ 2.10GHz |
| RAM | 128GB |
| GPU (number and type) | Two NVIDIA GeForce RTX3090 24G |
| CUDA version | 11.3 |
| Programming language | Python 3.9 |
| Deep learning framework | Pytorch (Torch 1.12.0, torchvision 0.13.0) |
| Specific dependencies | numpy, opencv-python, PyYAML, scipy, monai, scikit-image, edt, torch_scatter |
| Code | Cell-Segmenter |

## 2.4   Other Tricks

To improve generalizability on the unseen cells, a naive self-training is conducted over the complete unlabeled training set. All given unlabeled whole slide microscopes are cropped into $640 \times 640$ patches before pseudo-label generation. In practice, we find the noisy pseudo label may destroy the functionality of models on labeled data. Therefore, we oversample the labeled data to reduce the toxic gradients from pseudo-label during self-training. In inference time, sliding window inference and patch-wise NMS strategy are used to speed up inference time and reduce GPU memory consumption. Besides, the Omnipose is only adopted in grayscale microscopy.

# 3   Experiments

## 3.1   Dataset

In the training stage, we adopt the following public datasets: Cellpose, Omnipose. The two datasets are used to train Omnipose model to predict masks for cells with complex morphology. As for pretrained models, the ResNet-18 backbone on ImageNet [1] is adopted for all models.

## 3.2   Implementation Details

In this section, we will introduce the implementation details of our framework, including training environment, protocols and hyper-parameter settings.

### 3.2.1   Environment Settings

The development environments and requirements are presented in Table 1.

### 3.2.2   Training Protocols

**Data Augmentation**.   In the training stage, we apply random geometric transformation for all samples, including translation, scaling, rotation, shearing, and flipping. Meanwhile, we downsample or upsample cells based on the average diameter in the image, e.g., any image with an average cell diameter larger than 200px will be randomly downsampled to 0.125x, 0.25x, or 0.5x. Besides, the random gamma operation is adopted to make the model robust to bad light conditions. For brightfield microscopy images, we also adopt RandomColorJitter due to variations in the staining process.

**Patch Sampling**.   During training, we apply the random padded crop with size $640 \times 640$ to the transformed image and mask. Any boxes of cells with <20% pixels remaining will be omitted in the detection loss. For those unlabeled whole slide images, we crop them into $640 \times 640$ patches for acceleration. In the inference time, we adopt a $640 \times 640$ sliding window with an overlap factor of 0.125 and gaussian weights.

**Model Selection**.   We never use early stopping and always choose the model with the complete training.

**Ensemble Model Training**.   To improve time efficiency, models adopted in the pipeline are learned in a pretraining-finetuning way. After removing the noisy data in the training set, a baseline model is trained on the labeled data. Then pseudo labels of unlabeled data are generated. Finally, four branch models are trained with different settings. Here we give the main settings of our training process. For details of hyperparameter settings, please refer to config files.

Table 2: Training protocols for baseline model.

| Name | Settings |
| --- | --- |
| Network initialization | "he" normal initialization [18] |
| Batch size | 32 |
| Patch size | $3 \times 640 \times 640$ |
| Total epochs | 300 |
| Optimizer | SGD with nesterov momentum ($\mu = 0.9$) |
| Initial learning rate (lr) | 0.01 |
| Lr decay schedule | cosine annealing |
| Warmup | 3 epochs |
| Training time | around 80 hours |
| Loss function | CrossEntropy loss and YOLOv5 detection loss |
| Number of model parameters | 12.61M |
| Number of flops | 41.9G |

---

[1] https://download.pytorch.org/models/resnet18-5c106cde.pth

Table 3: Training protocols for general model finetuning. Unchanged parameters are not shown in this table.

| Name | Settings |
|---|---|
| Total epochs | 50 |
| Initial learning rate (lr) | 0.001 |
| Warmup | 1 epochs |
| Oversample | 8x - 30x |
| Training time | around 20 hours |

Table 4: Training protocols for Omnipose model. Unchanged parameters are not shown in this table.

| Name | Settings |
|---|---|
| Total epochs | 100 |
| Initial learning rate (lr) | 0.001 |
| Training time | around 10 hours |
| Loss function | Modified Omnipose loss |
| Number of model parameters | 12.61M |
| Number of flops | 41.9G |

## 4 Results and Discussion

In this section, we give the quantitive and qualitative experiment results on the TuningSet of the CellSeg contest. An efficiency evaluation is also conducted.

### 4.1 Quantitative Results on Tuning Set

As shown in Table 5, the F1-Score of the proposed framework on TuningSet reaches 0.8452, which verifies the effectiveness of our two-step watershed algorithm. Furthermore, the framework gains a margin of $0.85\%$ performance improvement after weakly training on each modality.

### 4.2 Qualitative Results on Tuning Set

Table 5: The ablation study of the proposed framework on TuningSet.

| Labeled | Two-step Watershed | Weakly | F1-score |
|---|---|---|---|
| ✓ | | | 0.8312 |
| ✓ | ✓ | | 0.8452 |
| ✓ | ✓ | ✓ | 0.8537 |

Figure 3 and 4 show some segmentation results of the AWF model in the proposed Cell Segmenter on NeurIPS 2022 Cell Segmentation TuningSet. From the two figures, we discover that our AWF is suitable for convex cells, regardless of their cluster degree. However, it has performs inferior in those clustered mutant cells, especially when bounding box centers of the cells are not in the cell body.

### 4.3 Segmentation Efficiency Results on Tuning Set

In this section, we plot the running time of the Cell Segmenter on the local workstation. As shown in Figure 5, most inference processes of images in TuningSet can finish in time tolerance. However, some of the images still slightly exceed the tolerance, possibly due to our local machine's low CPU frequency. Besides, for whole slide images (image id: 101), we find most of the running time (320.59 secs) is used in the watershed algorithm after non-max suppression ($> 80\%$) due to the single-threaded watershed implementation.

### 4.4 Results on Final Testing Set

Table 6 shows the performance of the proposed Cell Segmenter on the final test set. The weak performance of fluorescence modality may come from some potential bugs.

| Images | Energy | Cell Probability | Final Prediction |

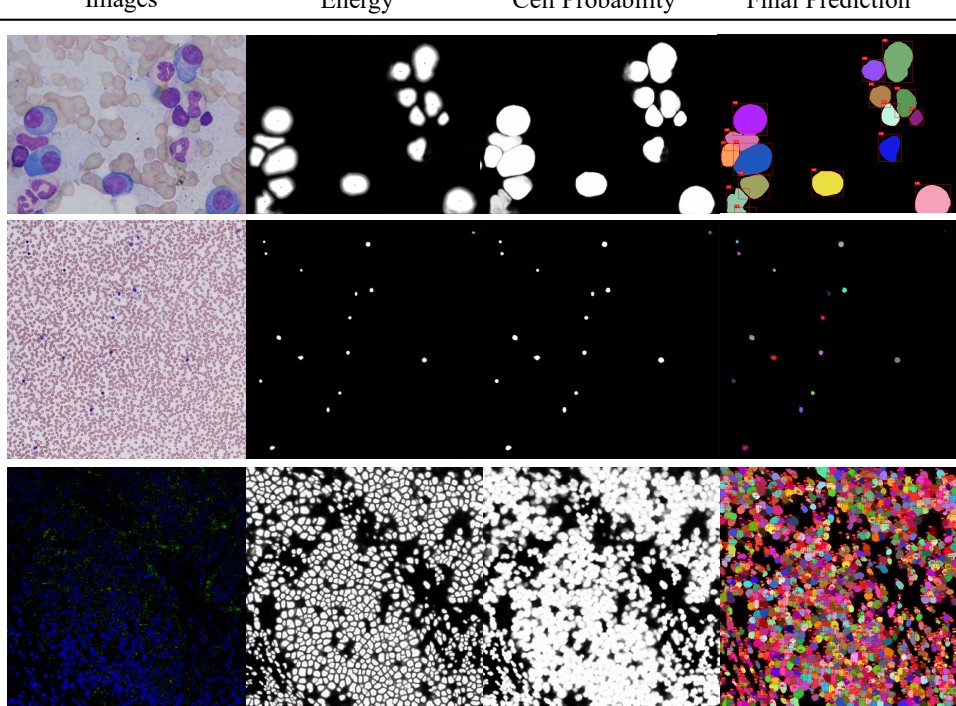

Figure 3: Prediction and corresponding intermediate representation maps of images with a good performance by the proposed AWF branch on the TuningSet. The 'Energy' column visualizes the watershed energy of input images, and the red points in the map represent centers of detection boxes before NMS operation.

| Images | Energy | Cell Probability | Final Prediction |

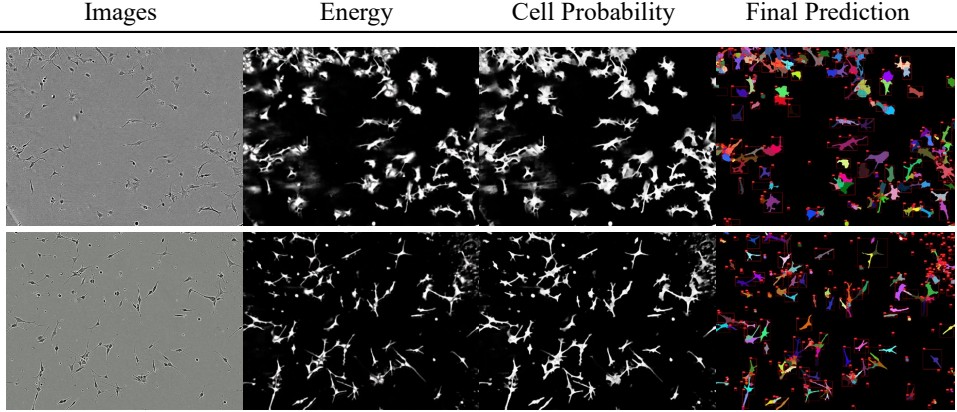

Figure 4: Prediction and corresponding intermediate representation maps of images with an inferior performance by the proposed AWF branch on the TuningSet. The 'Energy' column visualizes the watershed energy of input images, and the red points in the map represent centers of detection boxes before NMS operation.

Table 6: The quantitative results on the final test set.

| F1-Score | Modality | | | | |
| --- | --- | --- | --- | --- | --- |
| | Brightfield | DIC | Fluorescence | Phase Contrast | All |
| Median | 0.9009 | 0.6614 | 0.0492 | 0.8901 | 0.7759 |
| Mean | 0.8911 | 0.6025 | 0.1983 | 0.7849 | 0.6216 |

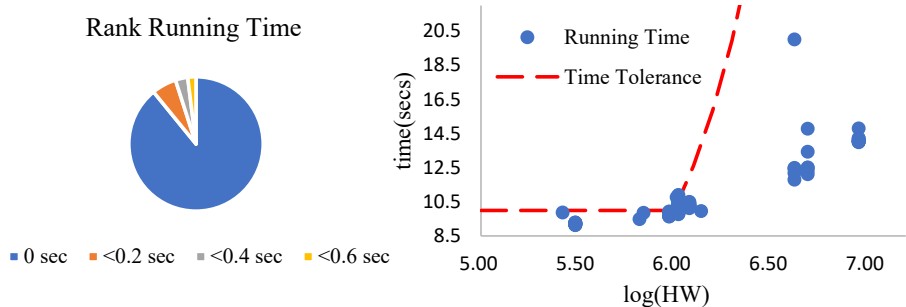

Figure 5: Time Consumption on TuningSet.

### 4.5 Limitation and Future Work

The proposed framework still contains some limitations: (i) We solve the general cell problem by adopting different methods on the different modalities, therefore the performance of the Cell Segmenter is strongly affected by the Quality Estimator; (ii) the Omnipose performs relatively poorly on large cells and is sensitive to post-processing; (iii) the NMS process for bounding boxes of mutant cells will lead to under-segmentation for some tilted cells. In the future, we will focus on the following directions: (i) morphology-independent marker synthesis for watershed segmentation based on detection algorithm; (ii) the more effective constraint of energy map; (iii) a quicker post-processing algorithm to reduce inference consumption.

## 5 Conclusion

In this paper, we propose a general cell segmentation framework for NeurIPS Weakly Supervised Cell Segmentation in Multi-modality High-Resolution Microscopy Images, named Cell Segmenter. To segment cells of arbitrary size and morphology, the whole pipeline automatically selects a suitable branch based on the output of the Quality Estimator. For those cells of simple or mutant shape, an Anchor-based Watershed Framework is used for size-irrelevant segmentation, while for those elongated cells, the recent Omnipose framework is applied. The F1-score of our framework reaches 0.8537 on CellSeg TuningSet and 0.6216 on the final test set, which verify the effectiveness of the proposed method.

## Acknowledgement

The authors of this paper declare that the segmentation method they implemented for participation in the NeurIPS 2022 Cell Segmentation challenge has not used any private datasets other than those provided by the organizers and the official external datasets and pretrained models. The proposed solution is fully automatic without any manual intervention.

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

# A    Appendix

## A.1    Network Architecture

### A.1.1    ResNet Layer, CSP BottleNeck, and SPPF

The ResNet Layer is a sequential container of Basic Residual Block, which is the same with torchvision's implementation.

The CSP BottleNeck, first proposed in [19], can be adopted to enhance the learning capability of CNN. The implementation of this module comes from the YOLOv5 official repository, and we apply it in the decoder architecture.

Spatial Pyramid Pooling [20] is an effective module to enhance the scale invariability of the CNN model. We adopt a fast implementation presented in the YOLOv5 official repo, which replaces multiscale pooling with iterative single-scale pooling.

### A.1.2    Network Details

Table 7 is the configurations of each layers in the two networks. The 'repeat' argument of module 'layer' specifies the basic block number in single layer.

Table 7: The architecture detail of AWF model. 'Conv' represents a nonlinear Conv-BatchNorm-SiLU block. 'C3' represents a CSP BottleNeck block. 'SegDetect' only contains two output Conv layers for detection and segmentation. The Omnipose model will replace the final 'SegDetect' module with a conv2d layer. For other detail of our model, please refer to the official implementation.

| Index | Module | Input | Arguments |
|---|---|---|---|
| 0 | Conv | -1 | in_channel=3, out_channel=64, kernel_size=7, stride=2, padding=3 |
| 1 | MaxPool2d | 0 | kernel_size=3, stride=2, padding=1 |
| 2 | Layer | 1 | in_channel=64, out_channel=64, repeat=2, stride=1 |
| 3 | Layer | 2 | in_channel=64, out_channel=128, repeat=2, stride=2 |
| 4 | Layer | 3 | in_channel=128, out_channel=256, repeat=2, stride=2 |
| 5 | Layer | 4 | in_channel=256, out_channel=512, repeat=2, stride=2 |
| 6 | SPPF | 5 | in_channel=512, out_channel=512, kernel_size=5 |
| 7 | Conv | 6 | in_channel=512, out_channel=256, kernel_size=1, stride=1 |
| 8 | Upsample | 7 | scale_factor=2, bilinear, align_corner |
| 9 | Concat | 8, 4 | dim=1 |
| 10 | C3 | 9 | in_channel=512, out_channel=256, shortcat=False |
| 11 | Conv | 10 | in_channel=256, out_channel=128, kernel_size=1, stride=1 |
| 12 | Upsample | 11 | scale_factor=2, bilinear, align_corner |
| 13 | Concat | 12, 3 | dim=1 |
| 14 | C3 | 13 | in_channel=256, out_channel=128, shortcat=False |
| 15 | Conv | 14 | in_channel=128, out_channel=64, kernel_size=3, stride=1, padding=1 |
| 16 | Upsample | 15 | scale_factor=2, bilinear, align_corner |
| 17 | Concat | 16, 2 | dim=1 |
| 18 | C3 | 17 | in_channel=128, out_channel=64, shortcat=False |
| 19 | Conv | 18 | in_channel=64, out_channel=64, kernel_size=3, stride=1, padding=1 |
| 20 | Upsample | 19 | scale_factor=2, bilinear, align_corner |
| 21 | Concat | 20, 0 | dim=1 |
| 22 | C3 | 21 | in_channel=128, out_channel=64, shortcat=False |
| 23 | Upsample | 22 | scale_factor=2, bilinear, align_corner |
| 24 | SegDetect | 14, 23 | det: in_channel=128, out_channel=6, seg: in_channel=64, out_channel=2 |

## A.2    The effectiveness of the Quality Estimator

In this section, we conduct a preliminary experiment to verify the effectiveness of the proposed Quality Estimation module. In detail, we first split the training/validation datasets into two parts: (i) better predictions with the AWF branch; (ii) better predictions with the Omnipose branch. Then the mean quality of the two data splits is calculated with the two branches, respectively. As shown in Table 8, there is a huge gap between the quality of the two branches on the 'Omni' split, which verifies the QE module's effectiveness.

Table 8: The mean prediction quality results of the two data splits predicted with the two models. The 'AWF' and 'Omni' in the second row mean the corresponding best model of the split. The leftmost column describes the used model. For example, '0.931' is the mean prediction quality of the 'Omni' split predicted with the Omnipose model on the training set.

| Quality | Training Set | | Tuning Set | |
|---|---|---|---|---|
| | AWF | Omni | AWF | Omni |
| AWF | 0.939 | 0.931 | 0.900 | 0.831 |
| Omni | 0.606 | 0.849 | 0.626 | 0.932 |

