# OpenReview forum: "Cell Segmenter: A General Framework for Multi-modality Cell Segmentation"
_NeurIPS.cc/2022/Challenge/CellSeg — Submitted to NeurIPS CellSeg 2022_

### Official Review · Reviewer_48KG · 2022-12-27
**The approach of quality estimation based on the cell's shape is interesting but needs further improvement.**

**Rating:** 8
**Confidence:** 4

**Review:**

## **Summary**

The authors proposed an All-Cell Segmenter framework to solve the multi-modality instance cell segmentation challenge. Based on the cell shape, they classified given cells into three types: simple cells, mutant cells, and elongated cells. They adopted the Anchor-based Watershed Framework(AWF), a detection-based watershed segmentation method for simple and mutant cells, and Omnipose for elongated cells. AWF combines YOLOv5 and watershed segmentation to achieve segmentation results optimized for high-convex cells. All-Cell Segmenter automatically performs quality estimation based on confidence to obtain better segmentation results between AWF and Omnipose. All-Cell Segmenter showed excellent performance quantitatively and qualitatively.

## **Pros**
1. The paper is well structured and contains the required contents of each section well enough.
2. The method of determining the postprocessor through quality estimation according to the cell's shape is meaningful.
3. An ablation study on the tuning set to confirm the effectiveness of the method is useful. Additionally, it is recommended to fill in the F1-Score of the combination of "labeled + weakly" for a more explicit ablation study.
## **Cons**
1. It is necessary to demonstrate theoretically or experimentally that quality estimation works reliably according to the cell shape.
2. Since the quality estimation process requires inference results of two AWFs and Omnipose branches, two inferences are required each time.
3. Omnipose prediction selection process is heuristic.
4. As mentioned by the authors, the performance of All-CellSegmenter depends on quality estimation, and it can cause performance degradation when there is a wrong quality estimation.
## **Minor**
1. In Abstract, F1-socre 0.8451 -> 0.8537
2. In Figure2, yolov5 -> YOLOv5

---

### Official Review · Reviewer_jnLF · 2022-12-28
**Paper well structured, small improvements needed**

**Rating:** 7
**Confidence:** 4

**Review:**

Summary:
The authors presented an all-purpose cell segmentation framework with two branches for Weakly Supervised Cell Segmentation in Multi-modality High-Resolution Microscopy Images. One branch is related to an anchor-based Watershed framework for those cells of simple or mutant shape, while the other is associated with the Omnipose framework for elongated cells. The framework can automatically select a suitable branch based on the results of the Quality Estimator. The final F1-score of the framework is 0.8537.

Pros:
The article is well constructed, and it is easy to follow.
The experiments are well conducted, and analysis is well performed.

Cons:
The Quality Estimator strongly affects the final results. Theorical and experimental proofs should be given for model reproducibility.
The Omnipose branch is sensitive to post-processing and does not perform stable on large cells. This part needs to be further improved. Apparently, the title “All-Cell Segmenter” is inappropriate.

---

### Decision · Program_Chairs · 2023-01-19

Accept